# Design and Development of Tools for Risk Evaluation of Diabetes and Cardiovascular Disease in Community Pharmacy

**DOI:** 10.3390/ijerph20042819

**Published:** 2023-02-05

**Authors:** Sarah Rondeaux, Tessa Braeckman, Mieke Beckwé, Dounia El Oueriaghli El Ghammaz, Dirk Devroey, Carine De Vriese

**Affiliations:** 1Department of Pharmacotherapy and Pharmaceutics, Faculty of Pharmacy, Université Libre de Bruxelles (ULB), 1050 Brussels, Belgium; 2Department of Family Medicine and Chronic Care, Faculty of Medicine, Vrije Universiteit Brussels (VUB), 1090 Brussels, Belgium

**Keywords:** community pharmacy services, patient education, diagnostic screening programs, diabetes mellitus, Belgium

## Abstract

There is an increasing motivation to implement pharmacist-led screening services in community pharmacies. This study aims to develop tools to support the pharmacist in the context of a diabetes and cardiovascular disease risk assessment service. Our development involved a multistep process using a user-centred approach, including a need assessment phase (14 patients, 17 pharmacists) and a creative design phase, followed by the evaluation of the materials (10 patients, 16 pharmacists). Three following themes covering educational needs emerged from stakeholders’ discussions: “content”, “layout”, and “form”, with three additional themes regarding the practical organisation: “software”, “awareness”, and “referral”. Based on the need assessment, tools for patient education purposes and awareness campaigns were created. During the development, special attention was paid to the writing style and structure with less text and more graphical colourful elements to suit patients with different health literacy and educational levels. The evaluation phase allowed researchers to observe participants engaging with the materials. Overall, participants were satisfied with the tools. The contents were considered valuable and relevant. However, adaptations were necessary to ensure their understanding and long-term usability. Finally, future research is required to evaluate the materials’ impact on patients’ behaviour towards their identified risk factors and ensure their effectiveness.

## 1. Introduction

Worldwide, healthcare systems are challenged with the rising rates of non-communicable chronic diseases (NCDs), such as diabetes and cardiovascular diseases, due to the ageing of the population, poor lifestyle behaviour and environmental factors. Their economic burden is projected to intensify in the future with devastating, long-term economic consequences for individuals and their households due to increased disabilities and premature death while being associated with heightened demand and expenditure of healthcare services and treatment [1]. This growing clinical and socio-economic burden greatly impacts people’s quality of life and consequently represents a significant challenge for optimal healthcare provision [2]. To lessen their impact, health systems need both individual and collective approaches that focus on reducing the modifiable risks associated with these diseases. Preventive strategies, such as early screening and chronic disease management services can help encourage individuals to take the actions necessary to achieve their healthcare goals. In that regard, health promotion, both to patients with or at risk of chronic diseases, in the form of tailored patient education, positively impacts the patient’s knowledge and understanding of the risks involved with their health status [3]. As such, community pharmacists, by their high accessibility, long opening hours, and the provision of pharmaceutical care without an appointment, are ideally placed for these services [4].

Recently, there has been increased interest in broadening the role of community pharmacists beyond the traditional medication dispenser to progressively shift to a provider of services and information [5,6]. Moreover, the increasing healthcare demands have led pharmacists to undertake more advanced clinical roles targeted to improve health outcomes, such as medication reviews, immunisations, health promotion, and patient education through patient and collaborative care [5,6]. Several systematic reviews have reported the positive impact of pharmacist interventions on patients’ therapeutic, safety and humanistic outcomes among different conditions, such as hypertension and diabetes mellitus [7,8,9,10]. These interventions usually involved medication review, education concerning health promotion, diseases and medication understanding, and specific risk reduction activities. Outcomes included a significantly improved quality of life, reduced disease risk, and premature mortality [7,8,9,10]. Therefore, pharmacists must now learn and possess specific knowledge and skill competencies to align with their new role of service provider along with other healthcare practitioners through collaborative practice.

In Belgium, the role of pharmacists is evolving more slowly, compared to their neighbouring countries despite an intrinsic motivation to play a more active role in primary care, chronic disease management and health promotion. Moreover, even if there is increasing research demonstrating the impact of pharmacist-led preventive services worldwide [11], evidence on how to provide a risk evaluation in community pharmacies is still limited. With that purpose, the Association of Pharmacists in Belgium (APB) conducted a six-month pilot project among a sample of pharmacies in the Brussels capital region. To facilitate the implementation and conduct of the pilot project, two universities collaborated to evaluate the feasibility of offering diabetes and cardiovascular disease screening services in community pharmacies. An implementation study was conducted to identify the contextual factors facilitating or hindering a successful implementation within community pharmacies and to explore the experiences of pharmacists and patients. Overall, the service was perceived as easy to perform and feasible in practice but will require external and financial support to guarantee its effectiveness and a sustainable, successful implementation [12].

Evidence shows that more vulnerable groups, such as the elderly and patients with chronic diseases, are most at risk of having insufficient health literacy while being the ones with the greatest healthcare needs and expenses [13,14]. Delivering services, such as a risk assessment, was an excellent opportunity for pharmacists to initiate a dialogue with their patients on their health and provide support in the management of their risk factors. Therefore, alongside the implementation study, the research team developed material and tools to support pharmacists in implementing the project and help them in counselling their patients, especially patients with limited health literacy. This paper aims to present an overview of the multistep process, using a user-centred approach, to design and develop tools for a diabetes and cardiovascular disease risk assessment service in community pharmacies.

## 2. Materials and Methods

### 2.1. Study Design

The development procedure was inspired by the basic steps for developing health education tools using a user-centred approach [11,12]. The development of the materials involved a multistep process, including a need assessment phase involving a literature review and discussions with stakeholders, and a creative design phase followed by the evaluation of the materials created. Figure 1 illustrates the different stages of the development process.

### 2.2. Need Assessment Phase

Initially, an assessment was undertaken to identify the needs of patients and pharmacists in the context of a risk evaluation in community pharmacies. First, a literature review and web research on the different tools regarding screening for diabetes and cardiovascular diseases were conducted. We intended to obtain information on the content, layout and variability of materials, questionnaires and tools to perform screening, or deliver patient education for the targeted diseases. Secondly, qualitative discussions with pharmacists and patients were organised to explore and expand on their needs by discussing the possible implementation of a risk evaluation service in community pharmacies and generating ideas using existing tools as a starting point.

Semi-structured focus groups were organised with pharmacists. Two methods were used for the recruitment. The researchers sent an email to pharmacists who are involved in internships with pharmacy students and the local pharmacists’ organisations to their members. To boost recruitment, researchers also delivered handouts in community pharmacies. During the focus groups, participants received a bundle of folders, existing tools and games illustrating possible topics and modes of presentation to elicit concrete opinions about content, layout, and usability. Following a joint discussion, participants prioritised the tools according to their preferences and needs.

Additionally, semi-structured individual phone interviews with patients were also carried out. Patients were recruited through an online post on a Facebook group and the patient’s community of a general practitioner via a WhatsApp group. The semi-structured interviews were conducted in French or Dutch, according to the patient’s preference, discussing their perception of the risk evaluation service and how they would like to receive counselling. Pharmacists and patients were eligible to participate if they were respectively practising and residing in the Brussels capital region and were able to speak, read, and understand French or Dutch. Patients also had to meet the inclusion criteria of the risk assessment pilot project (aged between 25–65 years old with no history of diabetes, cardiovascular or renal diseases). Data were collected from February to May 2020 until saturation, meaning no new information was collected during the last focus group and the last two interviews.

Semi-structured interviews and focus groups were audio recorded, transcribed ad-verbatim, and analysed using thematic analysis. Pharmacists’ and patients’ transcriptions were coded separately. Three researchers conducted the analysis and discussed every couple of interviews to converge nodes into an iterative coding tree and to ensure agreement on the conclusions. The data analysis was organised and managed using QSR International’s NVivo software 12 [15].

### 2.3. Creative Design and Development

Subsequently to the focus groups and interviews, the different tools and content to be created were defined. Considering the project resources before the launch of a pilot project simulating the risk evaluation in a sample of community pharmacies in the Brussels capital region, tools were selected to be developed into prototypes to answer the needs of pharmacists and patients.

Preliminary drafts were prepared in English for every tool and adjusted by experts for scientific validation. The drafts were translated into Dutch and French by the researchers. To validate the translation, a sample of 2 × 3 native speakers verified the language and grammatical correctness of the written text. The drafts were then sent to a translation company for an Arabic translation, as requested for additional language during the stakeholders’ discussions. Hereafter, they were adapted to designed prototypes by a graphic designer. In that way, the textual manuscripts were converted into visual prototypes. Additionally, a software developer from the Association of Pharmacists Belgium produced and designed a web tool to guide pharmacists during the risk evaluation.

### 2.4. Evaluation

In order to evaluate the acceptability and usability of the different tools, the creative design phase was followed by an implementation phase of the prototypes. The tools were printed, distributed, and used during the six months risk assessment pilot project within the community pharmacies. During the implementation study, semi-structured focus groups and face-to-face interviews with 16 participating pharmacists were organised to discuss the conduct of the project, including the use of the different tools by the pharmacists.

In addition to the implementation phase, the understanding and perception of the patient’s education materials were evaluated. Firstly, the readability index (RI), related to the level of schooling necessary to understand a given text, was assessed by the researchers using readability formulas on the French documents: the Simple Measure Of Gobbledygook (SMOG) [16], the Flesch–Kincaid grade level (FKGL) [17], the Coleman–Liau index (CLI) [18] and the FOG index. However, research shows that statistical tests of readability alone do not seem to be sufficient to ensure that information is understood [19]. Thereby, the understanding of the educational materials and the presentation of the information was evaluated with participants representative of the population targeted by the documents. Face-to-face interviews were conducted observing the users interacting with the prototypes. The interviews followed a semi-structured individual interview guide inspired by European Commission guidelines for the readability of the labels of medicines [20], the Gibson et al. questionnaire for the health education database [21] and the patient education materials assessment tool [22]. Questionnaires can be consulted in the Appendix A. The interview covered the general impression of the materials, user perception, comprehensibility of the contents, reader-friendliness, layout/design, length, and amount of illustrations.

All of the interviews were audio-recorded and transcribed ad verbatim. Pharmacists’ and patients’ transcriptions were coded separately and analysed to identify positive and negative perceptions, leading to adjustments.

## 3. Results

### 3.1. Need Assessment

There is considerable variability in the content and quality of diabetes and cardiovascular disease patient education materials [23,24]. Studies have shown that patient education material on diabetes and cardiovascular disease prevention should feature different themes, including basic knowledge of the diseases, their complications, self-monitoring, medication taking, and how to manage risk factors through healthy eating and physical activity [25,26]. However, studies have yet to define the ideal characteristics of a comprehensive patient education program in clinical practice [27].

During our qualitative study, three focus groups were conducted with 17 pharmacists, and 14 phone interviews were conducted with patients. Three themes covering educational needs emerged from the analysis of patients’ interviews and pharmacists’ focus groups: “content”, “layout”, and “form”. In addition, an additional theme regarding “training” emerged from the pharmacists’ focus groups. Details on the themes derived from the analysis and illustrative quotes can be found in Appendix B.

In general, patients and pharmacists agreed that information should address explanations of diabetes and cardiovascular diseases, their complications, their associated risk factors and lifestyle advice on preventing them. All groups found that the delivery of patient education was of utmost importance and should be individualised with concrete reachable objectives on a voluntary basis without being guilt-inducing. In that regard, training in motivational techniques was mentioned during the focus groups to help the patient achieve their goals, besides training in nutrition and multidisciplinary collaboration. Verbal explanation and counselling should be associated with visual aid and educative materials that the patients can take back home. Different layout possibilities were discussed. All agreed that educational material should be well-structured, reader-friendly and not too long. A popular recommendation was to use less text and more graphical and colourful elements.

Three additional themes emerged covering the practical organisation of the risk evaluation: “software”, “awareness”, and “referral”. Patients and pharmacists agreed that eye-catching awareness campaigns, including a set of items (e.g., flyers, animation videos, posters, etc.), should be developed to inform the target population of the possibility of being screened in community pharmacies. Additionally, developing software to record patients’ data for follow-up and to keep a schedule of appointments was also advised to facilitate the conduct of the risk evaluation. Finally, how to refer high-risk patients for a medical follow-up was discussed with pharmacists and patients. Even if the transfer of patients’ outcomes of the risk evaluation through an online platform to the general practitioners was preferred by pharmacists, mixed opinions were reported by the patients with a preference for a printout of their results.

### 3.2. Creative Design Phase and Development

Following the discussions with patients and pharmacists, we created a set of tools based on the assessed needs and project resources. Firstly, patient education materials were developed, including folders, a flip-over album, and a lifestyle advice booklet. The overview of the content of the patient education materials can be seen in Table 1.

The flip-over album was a patient education support intended to be displayed on the pharmacist’s counter and used as a visual aid to the pharmacist’s verbal counselling. On one side of the album, the patient could look at graphs, images, and pictograms to support the spoken words of the pharmacist, while the latter could look at the other side, which presented textual key information to be delivered. Two other documents to take back home to help patients who may benefit from written material were created. Folders were aimed to be distributed to patients with an identified risk of developing diabetes or cardiovascular diseases but contained only specific information focusing on the targeted diseases. However, a patient lifestyle booklet centred on a healthy lifestyle rather than disease management could be given to any patient. The booklet was divided into four following chapters with concrete pieces of advice on: “healthy eating”, “healthy body”, and “healthy mind” associated with a calendar to motivate patients to implement new healthy habits into their life.

During the development of the materials, special attention was paid to the writing style, trying to shorten words, substituting medical jargon for general terms, and using clear, direct, positive, personal, and conversational writing when possible. The active voice was also used to keep the text shorter and make it easier to read. Similarly, the design was carefully thought through using style and size of print that can influence the readability, colours, heading and bold writing to emphasise the important information.

In addition to the educative materials, a set of visibility tools was created to increase awareness of the risk evaluation program within the community pharmacies. To that purpose, flyers, posters, counter mats, and written questionnaires were delivered to pharmacists in two to four languages of their choice to put up and distribute according to their preferences. A handout to inform other practitioners on the purpose of the evaluation and write patients’ risk assessment results was developed. Finally, the Association of Belgian Pharmacies developed a web tool, available to the participating pharmacists exclusively online, to guide pharmacists through the risk evaluation questions, enter patients’ answers, and determine the assessed risk outcome for the targeted diseases. An electronic version of the educative materials and handout with their result could also be transferred by email at the patients’ request.

### 3.3. Evaluation

The readability tests, performed by the researchers, resulted in high variability in scores depending on the tests. None of our three patient education tools received a readability score between 7 and 8, as recommended by the National Institutes of Health (Figure 2). The folder on cardiovascular disease received the highest score throughout the tests, which means it would require a higher level of education to be easily read and thus is less accessible to the targeted population.

However, since the readability tests are developed for the English language, they were used as tools for information and were supplemented with an evaluation of the understanding of the target population, as recommended by the French National Authority for Health (HAS) [19]. Therefore, semi-structured interviews were conducted with 10 participants per material during the evaluation phase, which allowed us to identify misunderstood words and sentences, and adjust in consequences. An overview of participants’ characteristics can be seen in Table 2. An overview of the responses to the closed questions to the evaluation questionnaire can be found in Appendix C.

Participants had an overall positive impression of the educative materials, and the contents were considered useful and relevant. In particular, the illustrations of the diseases and risk factors were perceived as important information for laypeople and help participants in their understanding of the diseases. Patients appreciated the right balance between texts, illustrations, and moderate length, which was considered reader-friendly and thus maintained their eagerness to read. Moreover, the participants appreciated the reassuring and caring tone of the booklet. Suitable amounts of information were presented, and the contents were generally regarded as easy to understand and advice feasible in practice. Participants reported learning new information (e.g., nutriscore, nutrition advice) and ideas (e.g., new way to stay active, to manage stress) they could implement in their daily life.

The evaluation phase allowed researchers to observe how users read and manipulate the material. Thereby, they observed that the topics in the folders were not read in the order it was intended, which led to a reorganisation of the layout topics. This step also offered suggestions for modifications and improvements. Some issues focused on readability (minor adjustments in colours and font size), variability of representation (e.g., diversity in age groups or ethnicity) and minor adjustments in graphs and illustrations to improve understanding.

In addition, four focus groups and three face-to-face interviews with 16 pharmacists were organised to discuss the use of the tools during the risk assessment pilot project. An overview of pharmacists’ characteristics can be seen in Table 3.

Pharmacists appreciated the flip-over album as text-heavy presentations were avoided, and illustrations were considered valuable to explain complex conditions, such as diabetes. They were satisfied with the graphics and the layout, which were described as colourful and attractive. Pharmacists reported using the album during nearly every patient risk evaluation, but they often considered the album too long for personal use. Not every page was presented to the patients. However, the variability of use greatly depended on patient profiles and pharmacists, which made it difficult to reduce the number of pages significantly. To bypass this matter, a tab system to directly access specific pages to navigate more easily or removable pages to rearrange according to preferences was advised.

Visibility materials reportedly generated curiosity among some patients, which helped promote awareness of the project alongside the active recruitment approach by the pharmacists and could help reach more patients that would benefit from the risk evaluation.

Finally, the web tool helped the conduct of the risk evaluation and was greatly appreciated by the pharmacists as it allowed pharmacists to navigate through the risk evaluation questions and enter answers easily. Using a web tool was also perceived as efficient to optimise their time on counselling. Minor technical issues were reported. Other adjustments were discussed (e.g., adding a scheduling tool) with the APB and implemented if possible.

## 4. Discussion

This research complements a first study in Belgium examining the feasibility of offering a risk-assessment service for diabetes and cardiovascular diseases in community pharmacies and the factors influencing its implementation. Descriptive details on the methodology of the risk assessment, results of the implementation study and its feasibility in practice were reported by Rondeaux et al. [12]. Our present study aimed to develop materials to support pharmacists in their new role of providing a risk assessment service to their patients. Research has emphasised the importance of involving patients and stakeholders in the development of patient education aids to ensure their acceptability and avoid the need for future redesigns [29,30]. We have developed a set of tools taking into consideration the desirable characteristics of patient education materials [22] that can be used permanently and on a larger scale by incorporating users’ involvement through a multi-step process.

First, the literature for patient education on diabetes and cardiovascular diseases was explored. Secondly, we assessed the needs of pharmacists and patients and then developed a set of tools. Overall, the topics included in our material aligned with previous studies examining the educational needs and preferences of patients with diabetes and cardiovascular diseases [25,26]. However, they were limited to prevention purposes and did not include disease management topics as they did not correspond to the purpose of a screening service. Different items were created to fit the variability of the patient’s profile according to the determined risk.

A major Danish study of health literacy revealed that people with chronic conditions had more difficulty understanding health information and engaging with healthcare providers than the general population [31]. Furthermore, people with low levels of education had lower health literacy than people with high levels of education [31]. We intended to develop materials that would provide education to patients with different needs, health literacy levels, and learning abilities. Thus, special attention was given to shortening words, substituting jargon for general terms, illustrating complex concepts with well-designed pictures, and using a large font with headings and emphasis on important information, as recommended by other research and guidelines [32,33]. A previous study focusing on low literacy patient education material suggested evaluating the different prototypes with participants, which proved to be valuable in our research [34]. In the lifestyle booklet, the participants found the contents easy to understand, the amount of information useful, and acknowledged the right balance between text and illustrations to maintain attention and eagerness to read. However, minor changes were necessary to improve socio-cultural representations and understanding of graphs. Observing participants manipulate and read the folders during the evaluation allowed the researchers to understand their reading process and reorganise the topics in the intended reading order. The layout album was reported as a valuable visual aid to support pharmacists’ counselling. This format aligns with the evidence that people learn more deeply by combining words and pictures than by words alone [35]. However, if the content did not need any revision, adaptations to the format were necessary to facilitate its use.

The materials were designed with the intent to raise awareness of the risk factors and lead patients toward healthier behaviours. Pharmacists’ material was focused on teaching basic knowledge of the targeted diseases, while the patient’s materials focused more on practical strategies for behaviour changes. Short-term, easily implementable steps to work toward achieving long-term goals may offer daily incentives to change. Thus, a brief intervention based on social cognitive theory to teach patients how to develop their action plans could help them to create their own short-term goals for sustained behaviour change [36].

Moreover, pharmacists have shared that training in motivational interviewing could be beneficial to delivering patient education in the context of the screening service. Motivational interviewing is a collaborative, patient-centred communication skill that can facilitate behaviour change by stimulating a patient’s internal motivation [37]. Pharmacist-led motivational interviewing has been shown to be beneficial to patients’ adherence [38,39,40]. In a systematic review and meta-analysis of motivational interviewing in healthcare settings, 64% of the studies showed an effect of using motivational techniques even in brief encounters of 15 min. Nonetheless, more than one encounter with the patient seemed to ensure the effectiveness of motivational interviewing [41]. It is evident that all of the information on diabetes and cardiovascular diseases cannot be delivered during only one session. Pharmacists have shared that multiple sessions could help increase and motivate patients’ adherence to their short-term goals. Therefore, an education program could potentially be scheduled to address all of the issues specific to a patient, even if some topics would be discussed at every meeting while others less frequently.

Furthermore, if the education materials were developed in the context of screening in community pharmacies, the content was not specific to the setting. As such, the designed materials could be used and distributed in other clinical settings by other healthcare practitioners. The education could then involve different practitioners (e.g., medical practitioners, dieticians, tobacco cessation specialists, etc.) with specific abilities and experience to better care for the patient in a multidisciplinary approach.

Finally, this research included both French- and Dutch-speaking pharmacists and patients from the Brussels capital region. As such, the results reflect the perceptions and use of the tools in a metropolitan, densely populated area. Participation was also voluntary and thus may have included more highly motivated pharmacists and patients, making it, therefore not representative of all populations. Additional testing and research should be conducted in other regions and with a bigger sample of participants. Moreover, the materials were revised after carrying out the modifications from the evaluation phase. The latter was conducted in French with only an evaluation of the French materials. If commentary and suggestions relative to graphs, layouts, and pictures may be similar, additional evaluation should be conducted regarding understanding the materials in other languages. Furthermore, an assessment of the materials’ impact on patients’ behaviour in regard to the management of their risk factors was beyond the scope of this study. Following the distribution of the patient education materials, a formal assessment of knowledge improvement would gauge the success of communicating the intended information. It is also crucial to assess changes in self-efficacy, outcome expectations, and behaviour, as knowledge improvement is not always associated with improved health outcomes [42]. Thereby, further research is necessary to ensure their effectiveness. It is also essential to frequently revise the health education materials according to the latest scientific innovations and new knowledge in that field.

## 5. Conclusions

In Belgium, pharmacists could take up a more active role in identifying, screening, and counselling patients at risk of diabetes and cardiovascular diseases. Our study presented the development and design of materials to support pharmacists in this new role using a user-centred approach that can guide future development processes.

We developed different tools to fit the diverse needs and profiles of patients. First, a set of visibility materials to increase awareness of the risk evaluation. Secondly, patient education materials, such as a flip-over album with graphic illustrations to support the verbal counselling of the pharmacists, folders on the basic knowledge of cardiovascular diseases and diabetes and a booklet containing small reachable advice to help the patient implement healthier habits. Additionally, a web tool was developed by APB to guide the pharmacists through the multi-step risk evaluation. Overall, participants were satisfied with the tools created. The contents were perceived valuable and relevant in practice. However, future research is required to evaluate the materials’ impact on patients’ behaviour towards their identified risk factors and to ensure their effectiveness.

## Figures and Tables

**Figure 1 ijerph-20-02819-f001:**
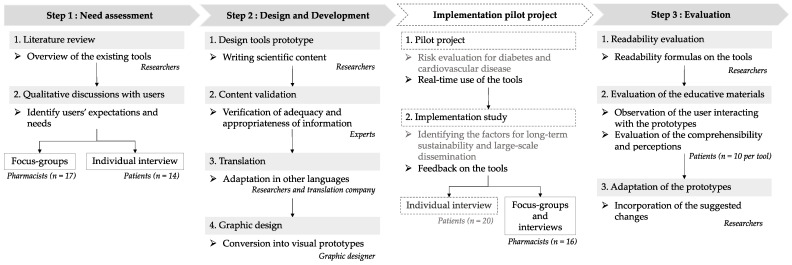
Overview of the different stages of the development and design process. Notes: Descriptive details on the methodology of the risk assessment project and the results of the implementation study (dotted lines in the graph) are presented in a separate article [12].

**Figure 2 ijerph-20-02819-f002:**
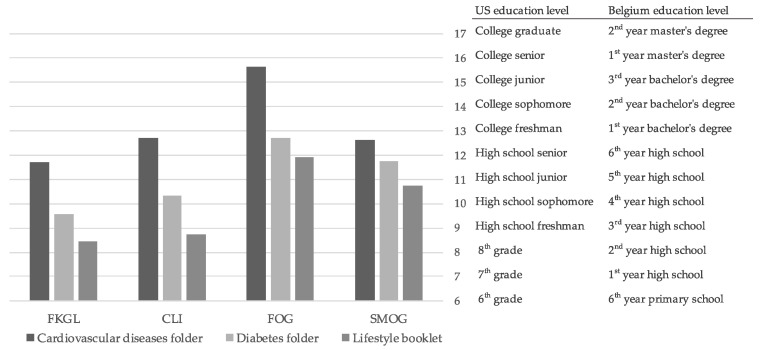
Readability evaluation of materials.

**Table 1 ijerph-20-02819-t001:** Topics covered in the patient education materials.

Flip over Album	Folder on Diabetes and Cardiovascular Disease	Lifestyle Patient Booklet
Protocol steps of the risk evaluation ▪ FINDRISC questionnaire ^1^ HbA1c result What is HbA1c? What is blood pressure? Cardiovascular evaluation Diabetes ▪ What is diabetes? Complications Cardiovascular disease ▪ What is high blood pressure? Complications Risk factors ▪ Overview Family history and ethnicity High blood sugar history Smoking Weight and waist circumference Lifestyle modifications: ▪ Food pyramid Glycaemic index table Physical activity triangle Energy expenses per activities How to reduce high blood pressure	Diabetes: ▪ Am I at risk? What is diabetes? Complications What can I do? Cardiovascular disease (CVDs): ▪ Am I at risk? What are CVDs? Complications What can I do?	Chapter 1: Healthy Eating ▪ What to prefer/limit/avoid How to make better choices 5 key rules to better eating Eating awareness Healthy habits Tips and tricks Chapter 2: Healthy Body ▪ Each step counts How to vary its activities Healthy habits Tips and tricks Chapter 3: Healthy mind ▪ Work on stress Time management Chapter 4: 28 days to better myself

Notes: ^1^ The diabetes risk evaluation was based on the combination of the Finnish diabetes risk score (FINDRISC) questionnaire [28], and a haemoglobin A1c reading.

**Table 2 ijerph-20-02819-t002:** Overview of the participants’ characteristics.

Characteristics	Folder Diabetes	Folder CVDs	Lifestyle Patient Booklet
Participants (n)	10	10	10
Male	3/10	4/10	5/10
Female	7/10	6/10	5/10
Age			
25–35 years	1/10	2/10	4/10
36–45 years	3/10	2/10	1/10
46–55 years	2/10	2/10	2/10
56–65 years	4/10	4/10	3/10
Education level			
Primary school	2/10	2/10	1/10
High school	3/10	2/10	4/10
Bachelor’s degree	3/10	4/10	4/10
Master’s degree	2/10	2/10	1/10
Interview duration range (mean)	11–62 (27 min)	8–25 (16 min)	9–42 (22 min)

**Table 3 ijerph-20-02819-t003:** Overview of the pharmacists’ characteristics.

Participants (n)	16
Male	6/16
Female	10/16
Age	
25–35 years	2/16
36–45 years	6/16
46–55 years	4/16
56–65 years	4/16

## Data Availability

Not applicable.

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
