# Peer review of "Design and Development of Tools for Risk Evaluation of Diabetes and Cardiovascular Disease in Community Pharmacy"

_ijerph, 2023, doi:10.3390/ijerph20042819_

Round 1

Reviewer 1 Report

This is a clearly written paper about a pertinent issue: to develop tools to support the pharmacist in the context of diabetes and cardiovascular diseases risk assessment service. The problem with the paper is that, in my judgment, it does not contribute anything new. On the one hand, it does not propose and defend any particular solution to the problem that it poses. On the other hand, it does not show (or at least, does not show adequately) how it is crucial for making use of the tools to support the pharmacist in the context of diabetes and cardiovascular diseases risk assessment service.  To have a definitive answer to the question of how the interests of humans are weighed against the interests of non-humans. The paper might, however, plausibly be interpreted as a paper providing an overview of the process to design and develop tools for risk evaluation. Furthermore, the paper lacks major steps that need to be followed to design and develop tools.

Author Response

Dear reviewer,

Thank you for allowing us to submit a revised draft of the manuscript “Design and Development of Tools for Risk Evaluation of Diabetes and Cardiovascular Disease in Community Pharmacy” for publication in this special issue of the International Journal of Environmental Research and Public Health. 

We appreciate the time and effort you and the reviewers assigned to provide feedback on our manuscript. 

Please find a point-by-point response to your comments and suggestions in the attached document.

Reviewer 2 Report

Greetings Authors 

Kindly find attached file and comply with amendments appear as comments on highlighted text. 

Author Response

Dear reviewer,

Thank you for giving us the opportunity to submit a revised draft of the manuscript “Design and Development of Tools for Risk Evaluation of Diabetes and Cardiovascular Disease in Community Pharmacy” for publication in this special issue of the International Journal of Environmental Research and Public Health.

We appreciate the time and effort you and other reviewers dedicated to providing feedback on our manuscript.

Please find a point-by-point response to your comments and suggestions in the attached document.

Reviewer 3 Report

Dear authors,

It was a pleasure to review this article of great importance from a public health perspective as well as pharmacists' involvement.

My suggestions and questions for improving the text are below.

Introduction

It is important to emphasize the economic burden of diabetes and cardiovascular diseases and the possible impact of the screening tools implementation in a few sentences.

Study design

 It would be most important to add a figure or another graphic to illustrate the complicated process described in the text. It was very hard to follow what patients did in the study and what participating pharmacists and what methods of evaluation were used at each step.

Materials and methods

Need assessment phase

It is unclear how many pharmacists responded to the call for research. What proportion of pharmacies in the area were contacted and how the list of the pharmacies was created? What proportion of pharmacists did respond in the approximation of reached pharmacists? This needs to be explained to get an impression of the representativeness of the sample size of 17 participating pharmacists.

This question is important for the patient's sample also. Did the WhatsApp group include just patients from one GP? What was the response rate in the approximation of the reached patients? Why were only patients aged 25-65 presented in the table of the results and questionnaires A i B. Inclusion criteria or you just got the answers from these age groups?

Creative design and development

Is there a reason why the first drafts were created in English and not in Dutch/French?

Did you validate the translation and how?

Results

Creative design phase and development

It is not clear what happens if a patient got the materials from the pharmacists. Did the pharmacists screen the patient at the moment or at some other moment when they attended the pharmacy? Did you measure how much time this screening and education last and is it practical to carry this out in a community pharmacy?

The software is mentioned in the text and it was not discussed how and if it was available to pharmacists in the pharmacies. What is evaluated by the pharmacists – no numerical data, scales of evaluation? Why just interview? In appendix A this is described as Software for schedule and follow-up. For me as the reader it is not clear do you use the software for screening eg. Questionnaires on diabetes etc. or just for schedule and follow-up. In what way?

Table 2.

The sample size should be written. I am not sure the data should be displayed this way because the sample is so small; it is not informative to see 70% as a number for female participants in folder diabetes when the group is 10 participants, it would be better to see 7/10 and not 70%. A separate column should be added – overall participants not separated by groups because of the 30 participants sample. Also, it needs to be clearly described are the groups (by folders) comparable by a categorical statistical test.

The basic characteristic of participating pharmacists is lacking – please add a table for this describing the sample. The age, and sex of pharmacist participants?

The sample size of participating pharmacists and patients should be mentioned in the abstract.

Discussion

This part needs to take into account limitations in the methodology, mainly the sample size of participants (just 17 pharmacists and 30 patients and the characteristics of patients – younger age). These limitations need to be discussed when explaining the result; do the participants represent the patients/pharmacists in the Brussels region or Belgium? If not is there a need to pilot on a bigger sample? Why the question of time duration was not measured or addressed – is it practical to use the screening tool in the pharmacy how much time does the pharmacist need to talk to the patient?

Best regards

Author Response

(The authors gave the same response as above.)

Round 2

Reviewer 1 Report

The authors have made almost all suggested changes and clarifications. I don't have nay more comments to suggestions to improve the manuscript. 

Reviewer 3 Report

Thank you for considering the suggestions and implementing the changes.